# NcIEMIL: Rethinking Decoupled Multiple Instance Learning Framework for Histopathological Slide Classification

Qiehe Sun[1]                                          SUNQH21@MAILS.TSINGHUA.EDU.CN
Doukou Jiang[2]                                              WOSHIJDK@126.COM
Jiawen Li[1]                                        LIJIAWEN21@MAILS.TSINGHUA.EDU.CN
Renao Yan[1]                                            YRA21@MAILS.TSINGHUA.EDU.CN
Yonghong He[*1]                                            HEYH@SZ.TSINGHUA.EDU.CN
Tian Guan[*1]                                          GUANTIAN@SZ.TSINGHUA.EDU.CN
Zhiqiang Cheng[*3]                              CHENGZHIQIANG2004@ALIYUN.COM

[1] *Shenzhen International Graduate School, Tsinghua University, China*

[2] *Department of Pathology, Shenzhen Center For Chronic Disease Control, China*

[3] *Department of Pathology, The Third People's Hospital of Shenzhen, China*

**Editors:** Accepted for publication at MIDL 2024

## Abstract

On account of superiority in annotation efficiency, multiple instance learning (MIL) has proved to be a promising framework for the whole slide image (WSI) classification in pathological diagnosis. However, current methods employ fully- or semi-decoupled frameworks to address the trade-off between billions of pixels and limited computational resources. This exacerbates the information bottleneck, leading to instance representations in a high-rank space that contains semantic redundancy compared to the potential low-rank category space of instances. Additionally, most negative instances are also independent of the positive properties of the bag. To address this, we introduce a weakly annotation-supervised filtering network, aiming to restore the low-rank nature of the slide-level representations. We then design a parallel aggregation structure that utilizes spatial attention mechanisms to model inter-correlation between instances and simultaneously assigns corresponding weights to channel dimensions to alleviate the redundant information introduced by feature extraction. Extensive experiments on the private gastrointestinal chemotaxis dataset and CAMELYON16 breast dataset show that our proposed framework is capable of handling both binary and multivariate classification problems and outperforms state-of-the-art MIL-based methods. The code is available at: https://github.com/polyethylene16/NcIEMIL.

**Keywords:** Multiple instance learning, Histopathological slide, Redundancy cleansing

## 1. Introduction

Histopathological slide examination which commonly requires a lot of time and effort from pathologists is seen as the "gold standard" in the clinical diagnosis (Aeffner et al., 2017; Cai et al., 2021). And computational pathology seeks to reduce the burden on physicians by employing algorithms (Kather et al., 2019; Skrede et al., 2020; Greenwald et al., 2022).

Tissue samples are stained, scanned, and stored as digital images with different magnifications in a pyramid structure known as the whole slide image (WSI). The lower layers of WSI are utilized to study the tumor heterogeneity in cell morphology, necessitating the

---

* Corresponding author

feeding of billions of pixels into networks at once. However, GPUs struggle to handle such many parameters simultaneously (Tellez et al., 2019). In this case, multiple instance learning (MIL) that decouples the computational process is more effective (Quellec et al., 2017).

Slides are meshed into numerous patches at a specific magnification. "patch" and "slide" correspond to "instance" and "bag" in MIL, respectively. A bag's characteristic is a collection of instance attributes. For slide classifcation, MIL frameworks are decoupled into two independent parts (Ilse et al., 2018), with a deep extractor being employed to understand salient features of instances, and a shallow aggregator responsible for the integration and mapping of bags. Since the above networks are trained separately, the computational cost is greatly reduced. Specifically, on the one hand, patching enables instances to be fed into the extractor in batches, avoiding the necessity to perform large matrix operations in a single pass. On the other hand, the global pooling layer at the end of the extractor substantially reduces dimensionality, facilitating straightforward linear transformations for the assembled bag-level representations. However, upon revisiting the aforementioned process, we identified information redundancy originating from multiple sources:

- **Gradient operation and backpropagation.** Slide-level labels serve as the exclusive source of supervised signals, and their gradients are solely fed back into the shallow aggregator. Conversely, the extractor is not constrained, resulting in a lack of understanding regarding the task-relevant property of the instances.

- **Instance-level feature concatenation.** In slides, the distribution of key instances tends to be sparse. However, when forming a bag embedding, all instances are involved, which leads to many irrelevant instances being of interest during aggregation.

- **Redundancy in the channel dimension.** For a given task, instances can be perceived as manifolds in a low-dimensional space. However, extractors often embed instances in high-dimension to prevent information loss, regardless of their utility.

For this reason, we sought to optimize the interference terms mentioned above and proposed a Non-crucial Information Elimination-based MIL (NcIEMIL) architecture. In customizing the extractor, we followed the approach of Campanella et al. (Campanella et al., 2019) by associating the corresponding bag-level annotation with the single instance that the extractor deems most likely to be positive, thus making full use of limited supervised signals. Subsequently, we exclude irrelevant instances based on the ranking of positive probability and forming a more refined bag embedding. The noise of channel dimension is specifically addressed during aggregation. We design a bi-parallel aggregator, introducing channel attention to weigh each embedding dimension, aiming to approximate the low-dimensional manifold. Simultaneously, we maintain the advantage of spatial attention, dynamically modeling the influence of different instances. To comprehensively validate the framework's effectiveness, we conducted experiments on the CAMELYON16 dataset and the retrospectively collected BgIM gastric mucosal biopsy dataset.

## 2. Related Work

Initially employed for drug activity prediction (Dietterich et al., 1997), multiple instance learning is widely used in the whole slide image research (Campanella et al., 2019; Kanavati

et al., 2020; Chen et al., 2021a; Marini et al., 2022). It aligns with the characteristics of pathology data, which is large but unlabeled, intending to reduce the daily workload of pathologists. MIL is categorized into instance- and embedding-level methods (Ilse et al., 2018). Extensive research has been conducted on the latter as it is well-suited for more complex application scenarios, and our method also belongs to this category.

### 2.1. Feature Extraction

Due to the absence of a standard database, MIL extractors typically employ neural networks pre-trained on ImageNet (Deng et al., 2009) to map instances into a hidden space, retaining basic knowledge of color and texture (Lu et al., 2021; Shao et al., 2021; Zhang et al., 2022). However, histopathological images significantly differ from natural images (Wang et al., 2022; Kang et al., 2023), a distinction increasingly emphasized with the advancement of self-supervised learning and extended to MIL (Li et al., 2021; Chen et al., 2022). Additionally, the semi-decoupled framework selects instances deemed important by the aggregator (Qu et al., 2022; Yu et al., 2023), providing supervision to the extractor but being prone to introducing errors. In contrast, we fully utilize slide-level labels, referencing instance-level methods (Campanella et al., 2019) to impart more determinism to the extraction network.

### 2.2. Instance Filtering

To identify the active region and eliminate extraneous interference, the instances were manually recalibrated by introducing extra supervision (Wang et al., 2019). Class activation mapping (CAM) (Zhou et al., 2016) was used to indicate lesion regions on thumbnails of WSI, although with some imprecision (Chen et al., 2021b). In this paper, we calculate the positive scores and select discriminative instances accordingly.

### 2.3. Attention-based Aggregation

Max- and avg-pooling serve as the simplest aggregation functions, representing two extreme assumptions in the MIL formulation that are prone to opposite judgments due to biases in the extraction process. Recognizing the diverse contributions of each instance, attention mechanism is employed to efficiently aggregate representations of instances (Ilse et al., 2018; Lu et al., 2021). Due to the great success of Transformer (Vaswani et al., 2017), self-attention (SA) has emerged as an alternative (Shao et al., 2021; Zheng et al., 2022). However, its quadratic complexity is called into question due to the significantly larger number of tokens in MIL compared to other domains. In this paper, by eliminating most non-discriminative instances, our aggregator is still constructed on top of SA, maintaining computational efficiency within an acceptable range.

## 3. Methodology

### 3.1. Multiple Instance Learning Formulation

For clarity, we present the formulation of MIL. Considering a binary problem as an example, for a given bag $\mathbf{X} = \{\mathbf{x}_1, \mathbf{x}_2, \cdots, \mathbf{x}_n\}$ and its corresponding label $Y \in \{0, 1\}$, where $\mathbf{x}_i$

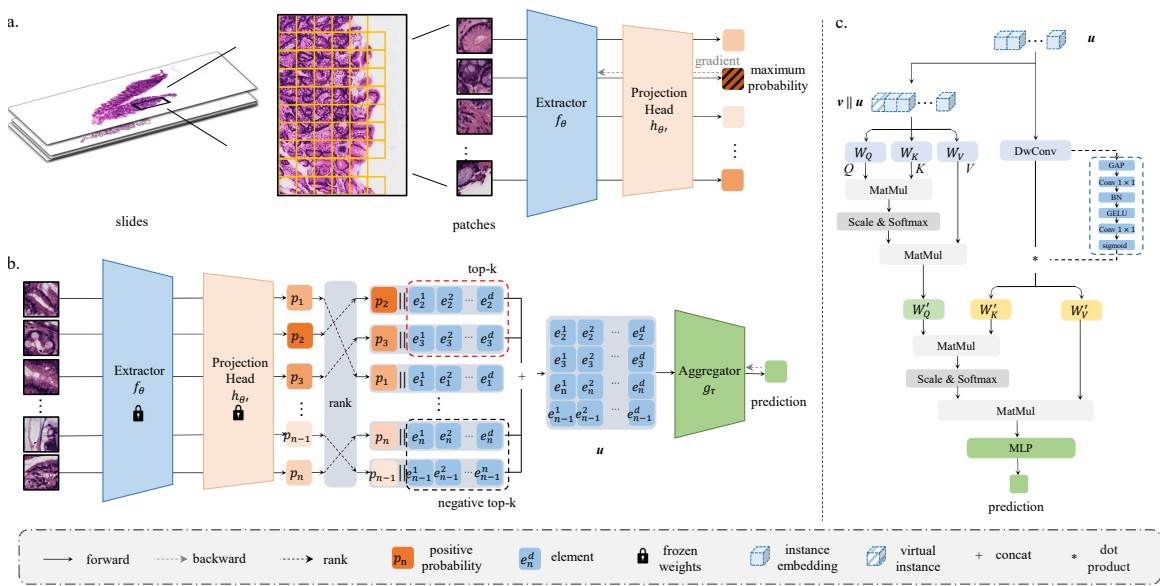

Figure 1: Overview of NcIEMIL. (a) Instances are fed into the extractor and projection head on a bag basis, and the instance with the highest probability is involved as a representative of its corresponding bag. (b) Instances within a bag are re-ranked, and bi-directional sampling is employed to generate bag embeddings. (c) A hybrid attention-based aggregator, comprising parallel spatial and channel attention mechanisms, is employed to mitigate information redundancy.

denotes the $i$-th instance of $\mathbf{X}$ and $n$ is the numbr of instances. We assume that the true label of $\mathbf{x}_i$ is $y_i \in \{0, 1\}$, which is not known in practice. Then MIL can be described as:

$$Y = \begin{cases} 0, & \text{iff } \sum_i y_i = 0, \\ 1, & \text{otherwise} \end{cases} \tag{1}$$

Assuming the instances within $\mathbf{X}$ are independently and identically distributed, Eq. 1 can be generalized (Zaheer et al., 2017) as:

$$Y = S(\mathbf{X}) = g(\textstyle\sum_i f(\mathbf{x}_i)) \tag{2}$$

where $S(\cdot)$ is a scoring function, and $f(\cdot)$ and $g(\cdot)$ are two suitable transformations. MIL is decoupled into two steps. In the embedding-level approach, $f(\cdot)$ is responsible for feature embedding. The summation function $\sum$ transforms into concatenation, and $g(\cdot)$ is an aggregation function that maps the bag-level representation to the category space.

### 3.2. Non-crucial Information Elimination-based MIL

In practice, two different networks, $f_\theta$ and $g_\tau$, perform the duties of $f(\cdot)$ and $g(\cdot)$, where $\theta$ and $\tau$ are learnable parameters. We observed that a substantial amount of irrelevant

information was forced to be retained or introduced during the process. Hence, we introduce **N**on-**c**rucial **I**nformation **E**limination (NcIE) to alleviate this trend, as shown in Fig. 1.

**Weakly-supervision for Extraction.** The gradient backpropagation stops between the aggregator and the extractor due to physical patching and splicing. As a result, the extractor remains unconstrained. Taking inspiration from the work of Campanella et al. (Campanella et al., 2019), we incorporate the ground truth $Y$ into the pre-training of the extractor. In particular, we append a mapping head $h_{\theta'}$ to $f_\theta$ to generate the positive probability $P(\mathbf{x}_i) = \mathbb{P}(h_{\theta'}(f_\theta(\mathbf{x}_i)) = y^* | \mathbf{x}_i)$ for the instance $\mathbf{x}_i$, where $y^*$ represents the positive class. The instance with the highest positive probability contributes to the parameter update, and the loss can be expressed as:

$$\mathcal{L} = -Y\log(\hat{p}) - (1 - Y)\log(1 - \hat{p}), \quad \hat{p} = max\{P(\mathbf{x}_i)|i = 1, 2, \cdots, n\} \tag{3}$$

where $\hat{y}$ is the score of the instance that is most likely to be positive. $n$ is not a constant as the foreground areas of slides vary, and $\theta$ and $\theta'$ are updated simultaneously.

**Discriminative Instance Selection.** In histopathological sections, only a small portion is pertinent to the problem. Numerous irrelevant instances introduce noise for the aggregator, leading to a drastic shift in attention (Yan et al., 2023). We precisely filter potential discriminative instances by assigning each instance from the same bag a corresponding score that is produced by $h_{\theta'}$, and then re-ranking instances in descending order:

$$\mathbf{X} = \{\mathbf{x}'_j | P(\mathbf{x}'_1) \geq P(\mathbf{x}'_2) \geq \cdots \geq P(\mathbf{x}'_n)\} \tag{4}$$

The first $K$ and the last $K$ instances of $\mathbf{X}$ are chosen to create a new pseudo-bag $\mathbf{X}' = \{\mathbf{x}'_1, \cdots, \mathbf{x}'_K, \mathbf{x}'_{n-K+1}, \cdots, \mathbf{x}'_n\}$. Including the latter $K$ instances aims to implicitly construct negative samples for discriminative instances while avoiding extreme data distribution.

**Hybrid Attention-Based Aggregator.** According to Eq. 1, for a given task, ideally, $f(\mathbf{x}_i)$ should be a low-dimensional manifold indicative of potential categories. However, $f_\theta(\mathbf{x}_i)$ is commonly a high-dimensional vector, implying redundancy in the channel dimension of bag embedding. Therefore, we use squeeze and excitation (Hu et al., 2018) to capture channel validity for it dynamically:

$$\mathbf{u}_{channel} = \text{sigmoid}(\mathbf{W}_2\sigma(\mathbf{W}_1 \cdot \frac{1}{2K}\sum_{k=1}^{2K}\mathbf{u}(k))) \cdot \mathbf{u} \tag{5}$$

where $\mathbf{u} = ||_{\mathbf{x}_k \in \mathbf{X}'}f_\theta(\mathbf{x}_k) \in \mathbb{R}^{2K \times d}$ denotes the bag embedding, obtained by concatenating instance embeddings, with $d$ being the embedding dimension. $\sigma$ is an activation function, while $\mathbf{W}_1$ and $\mathbf{W}_2$ are linear transformations. We maintain self-attention to model the influence of instances and their spatial correlation:

$$\begin{cases} \mathbf{u}_{spatial} = \text{softmax}(\mathbf{Q}\mathbf{K}^T/\sqrt{d})\mathbf{V} \\ \mathbf{Q} = (\mathbf{v}||\mathbf{u})\mathbf{W_Q}, \mathbf{K} = (\mathbf{v}||\mathbf{u})\mathbf{W_K}, \mathbf{V} = (\mathbf{v}||\mathbf{u})\mathbf{W_V} \end{cases} \tag{6}$$

where $\mathbf{W_Q}$, $\mathbf{W_K}$ and $\mathbf{W_V}$ are learnable, and $\mathbf{v} \in \mathbb{R}^{1 \times d}$ is a learnable embedding that represents a virtual instance used for classification. The above process is illustrated in Fig.

Table 1: Details of BgIM dataset division.

| | Severity of Intestinalization | | | |
|---|---|---|---|---|
| | - | + | ++ | +++ |
| Train | 56 | 74 | 32 | 16 |
| Test | 13 | 18 | 8 | 3 |

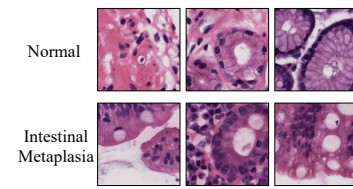

Figure 2: Several samples from BgIM.

1 (c). We designed an aggregator based on hybrid attention in a "parallel-merge" structure. The intersection of attention is achieved through cross-attention:

$$\mathbf{u}' = \text{softmax}(\mathbf{u}_{spatial}\mathbf{W}'_{\mathbf{Q}}(\mathbf{u}_{channel}\mathbf{W}'_{\mathbf{K}})^T/\sqrt{d})\mathbf{u}_{channel}\mathbf{W}'_{\mathbf{V}} \qquad (7)$$

where $\mathbf{W}'_{\mathbf{Q}}$, $\mathbf{W}'_{\mathbf{K}}$ and $\mathbf{W}'_{\mathbf{V}}$ are also learnable linear projections. Then the aggregated $\mathbf{v}'$ is extracted from the spatial dimension of $\mathbf{u}' \in \mathbb{R}^{(1+2K)\times d}$ and converted to the result by a fully-connected layer.

## 4. Experiment

### 4.1. Datasets

**CAMELYON16 Breast Dataset.** Camelyon16 (Bejnordi et al., 2017) is a publicly unbalanced dataset focused on differentiating between cancer and non-cancer cases for metastasis detection in breast cancer, which consists of 270 slides for training and 129 for testing. Following pre-processing, we acquired approximately 2.6 million patches at $20\times$ magnification, averaging 7,346 patches per slide.

**BgIM Gastric Mucosa Dataset**. Intestinal epithelial metaplasia is a common lesion of the gastric mucosa, occurring in many chronic gastric diseases, and is typically considered a precancerous condition (Correa et al., 2010). We retrospectively gathered 220 gastric mucosal biopsies and assessed the 4-grading results for intestinal epithelial metaplasia with two pathologists. On average, 1,320 patches were extracted per slide at $40\times$ magnification. More details are shown in Tab. 1 and Fig. 2.

### 4.2. Experiment Setup and Evaluation Metrics

We used the OTSU algorithm to create foreground masks for all WSIs, producing non-overlapping $256 \times 256$ patches. For CAMELYON16, we performed a 5-fold cross-validation on the official training set and reported the average performance on the official test set. We artificially divided the BgIM dataset into an 8:2 ratio for training and testing, respectively. We report the average performance across five different sets of random seeds. Metrics used include accuracy (ACC), area under the curve (AUC), and F1-score.

### 4.3. Implementation Details

We used swin-tiny (Liu et al., 2021) as the backbone of the extractor. When pre-training, a projection head was added at its end to obtain a positive probability, and a batch size of

128 was employed. In the aggregation phase, the bag embedding was increased from 768 to 1024 dimensions using a fully connected layer. Because the number of focus instances was fixed at $2K$ ($K = 512$ for CAMELYON16 and $K = 128$ for BgIM), the mini-batch was set to 4, differing from 1 in other comparison methods. We chose SGD as the optimizer with a learning rate of 2e-4 and weight decay of 1e-5, and the loss function used was cross-entropy. All experiments were conducted on an RTX 3090.

Table 2: **Comparative results on Camelyon16, and BgIM.** Subscripts indicate the standard deviation of the 5-fold cross-validation. The best performance is then marked in bold.

| Method | CAMELYON16 | | | BgIM | | |
|---|---|---|---|---|---|---|
| | ACC | AUC | F1-Score | ACC | AUC | F1-Score |
| Naive | $62.40_{1.42}$ | $62.16_{1.40}$ | $45.15_{5.28}$ | $52.98_{3.32}$ | $76.58_{1.28}$ | $44.27_{4.44}$ |
| Fully-supervised | $91.32_{1.99}$ | $94.93_{0.87}$ | $90.73_{2.12}$ | $91.91_{1.17}$ | $98.72_{0.33}$ | $91.57_{2.61}$ |
| MIL-RNN | $82.64_{2.17}$ | $85.02_{2.73}$ | $80.64_{1.94}$ | $53.81_{3.56}$ | $80.19_{1.86}$ | $48.55_{2.81}$ |
| ABMIL | $82.17_{1.77}$ | $84.05_{2.27}$ | $80.21_{1.66}$ | $60.00_{0.95}$ | $86.92_{0.46}$ | $51.13_{0.15}$ |
| CLAM-MB | $82.48_{3.27}$ | $81.20_{2.76}$ | $80.81_{3.37}$ | $64.70_{4.88}$ | $86.32_{2.69}$ | $56.50_{2.12}$ |
| DSMIL | $81.86_{1.44}$ | $79.71_{2.87}$ | $79.31_{1.31}$ | $60.00_{3.50}$ | $85.38_{0.77}$ | $52.08_{3.28}$ |
| TransMIL | $84.81_{1.60}$ | $88.13_{1.46}$ | $83.32_{1.17}$ | $73.81_{3.37}$ | $90.65_{0.70}$ | $64.9_{3.26}$ |
| ILRA-MIL | $84.65_{2.93}$ | $85.42_{2.24}$ | $82.90_{3.07}$ | $75.24_{2.43}$ | $91.28_{0.84}$ | $66.76_{2.95}$ |
| NcIEMIL | $\mathbf{86.05_{1.55}}$ | $\mathbf{89.68_{2.10}}$ | $\mathbf{85.26_{1.54}}$ | $\mathbf{85.23_{0.95}}$ | $\mathbf{95.87_{0.60}}$ | $\mathbf{81.20_{0.94}}$ |

### 4.4. Quantitative Results

Our NcIEMIL was compared with MIL-RNN (Campanella et al., 2019), ABMIL (Ilse et al., 2018), CLAM (Lu et al., 2021), DSMIL (Li et al., 2021), TransMIL (Shao et al., 2021), and ILRA-MIL (Xiang and Zhang, 2022). Tab. 2 presents the results on the two datasets. For CAMELYON16, our model shows improvements of 1.24%, 1.55%, and 1.94% in ACC, AUC, and F1-score, respectively, compared to the optimal model. While in the cumulative grading task with BgIM, our model exhibits greater effectiveness, surpassing the most effective ILRA-MIL by 9.99%, 4.59%, and 14.44% in ACC, AUC, and F1-score, respectively. Additionally, we reported significance verification to indicate that our method is significantly better than the baseline methods, as shown in appendix A.

Furthermore, we introduce two non-MIL methods: navie and fully-supervised approaches. The naive image-wise approach involved directly classifying downsampled WSI thumbnails using a swin-tiny network, akin to the approach in (Chen et al., 2021a). The fully-supervised approach used a batch of patches with real labels to train a patch-level swin-tiny network. The results of the slides were obtained by the mean value of all instances. The results show that the proposed method captures slice image details and performs much better than the naive method, but is slightly inferior to the fully supervised method due to the extractor's lack of factual knowledge. To demonstrate the accuracy of the discriminative instance selection, we also performed the corresponding visualizations, as shown in appendix B.

### 4.5. Ablation Studies

Ablation results are presented in Tab. 3. Initially, we validated the effectiveness of channel attention by removing the blue box in Fig. 1(c). The results demonstrate that, although modest, channel attention does enhance performance. Subsequently, we modified the way of selecting focus instances, replacing bi-directional sampling with single-directional sampling and random sampling (single sampling and random sampling in Tab. 3). The results indicated that bi-directional sampling significantly outperformed the other two approaches, validating our hypotheses regarding data distribution and implicit contrasts in bag embedding. We then tested different extractors, including replacing the original one with a network pre-trained on ImageNet and pre-trained by self-supervised learning (Wang et al., 2022). The selection of focal instances remains unchanged. Results show that task-specific relevant weakly supervised training of the extractor is effective. We also performed ablation exper-

Table 3: **Ablation Studies on Camelyon16, and BgIM.** Subscripts indicate the standard deviation of the 5-fold cross-validation. The best performance is then marked in bold.

| Ablation item | CAMELYON16 | | | BgIM | | |
|---|---|---|---|---|---|---|
| | ACC | AUC | F1-score | ACC | AUC | F1-score |
| w/o channel attention | $85.89_{1.50}$ | $88.86_{1.80}$ | $84.86_{1.47}$ | $84.28_{1.17}$ | $95.86_{0.65}$ | $79.62_{1.53}$ |
| w/ random sampling | $83.72_{1.90}$ | $85.80_{3.24}$ | $81.76_{2.31}$ | $80.95_{1.51}$ | $94.50_{0.62}$ | $78.07_{1.14}$ |
| w/ single sampling | $85.43_{1.98}$ | $87.17_{3.34}$ | $\mathbf{86.13_{1.69}}$ | $80.47_{0.95}$ | $94.00_{0.49}$ | $72.79_{2.45}$ |
| w/ ImageNet weight | $85.12_{1.58}$ | $87.02_{3.87}$ | $84.13_{1.22}$ | $80.00_{1.90}$ | $92.49_{0.97}$ | $70.41_{1.05}$ |
| w/ ctranspath weight | $85.73_{1.69}$ | $88.45_{1.61}$ | $84.84_{1.64}$ | $77.62_{4.15}$ | $92.70_{0.66}$ | $70.16_{5.87}$ |
| w/ small $K$ | $85.75_{1.84}$ | $\mathbf{90.06_{1.87}}$ | $84.87_{1.68}$ | $83.33_{1.51}$ | $95.21_{0.50}$ | $80.09_{2.14}$ |
| w/ medium $K$ | $85.36_{1.49}$ | $89.46_{1.30}$ | $84.01_{1.30}$ | $84.28_{1.90}$ | $94.90_{0.83}$ | $80.00_{1.95}$ |
| NcIEMIL | $\mathbf{86.05_{1.55}}$ | $89.68_{2.10}$ | $85.26_{1.54}$ | $\mathbf{85.23_{0.95}}$ | $\mathbf{95.87_{0.60}}$ | $\mathbf{81.20_{0.94}}$ |

iments on $K$. We introduced two values for $K$: a small $K$ ($K = 128$ for CAMELYON16 and $K = 32$ for BgIM) and a medium $K$ ($K = 288$ for CAMELYON16 and $K = 72$ for BgIM), and assessed their impacts on the results. The results show that a larger $K$ has more advantages but is limited by the slide area.

## 5. Conclusion

In this paper, we reconsider the decoupled MIL framework and assess the noise sources. To reduce information redundancy, we reshape the focal instance selection by employing weakly supervised training extractors and then create a hybrid attention-based aggregator. We collected a gastric mucosal biopsy dataset, BgIM, to validate the method's effectiveness. Extensive experiments on CAMELYON16 and BgIM demonstrate that our method's performance is comparable to the state-of-the-art. However, the approach still has shortcomings. For instance, the weakly supervised training of aggregators tends to bring positive instances with different semantic information closer in the feature space. Therefore, optimizing this process is a consideration for future work.

## Acknowledgments

We express our gratitude to Shenzhen First People's Hospital for providing de-identified data. The work was supported in part by the Development and Reform Commission of Shenzhen Municipality (Number: XMHT20230115004, KCXFZ20201221173207022). The authors declare that they have no known competing financial interests or personal relationships that could have appeared to influence the work reported in this paper.

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

## Appendix A. Significance Verification

We reported the p-values of two-sample t-tests on ACC, AUC, and F1-Score between our method and baselines respectively. $p < 0.05$ indicates that our method is significantly better than the baseline method. The results show that for the CAMELYON16 dataset,

Table 4: Significance analysis results on CAMELYON16 and BgIM datasets.

| Ablation item | CAMELYON16 | | | BgIM | | |
|---|---|---|---|---|---|---|
| | ACC | AUC | F1-score | ACC | AUC | F1-score |
| MIL-RNN | 0.0086 | 0.0194 | 0.0006 | $5.25e-5$ | 0.0001 | $1.91e-5$ |
| ABMIL | 0.0011 | 0.0060 | 0.0017 | $8.51e-7$ | $7.48e-5$ | $1.95e-7$ |
| CLAM-MB | 0.0318 | 0.0089 | 0.0193 | 0.0008 | 0.0008 | $3.09e-5$ |
| DSMIL | 0.0086 | 0.0059 | 0.0015 | $4.92e-5$ | 0.0001 | $4.63e-5$ |
| TransMIL | 0.1091 | 0.1874 | 0.0451 | 0.0010 | 0.0003 | 0.0005 |
| ILRA-MIL | 0.1893 | 0.0581 | 0.1191 | 0.0002 | 0.0012 | 0.0002 |

our method significantly outperforms all baseline methods except TransMIL and ILRA-MIL. For the BgIM dataset, our method significantly outperforms all baseline methods, as shown in the Tab. 4.

## Appendix B. Visualization

We present the top-3 instances and the negative top-3 instances from the 4-grades slides of BgIM and normal, micro-, and macro-metastasis slides of CAMELYON16, respectively. For CAMELYON16, the projection head trained with weak supervision identifies the patch containing the cancerous region. This holds true for both micro- and macro-metastatic cancer slides, as shown in Fig. 3. And in the BgIM dataset, for slides with intestinal

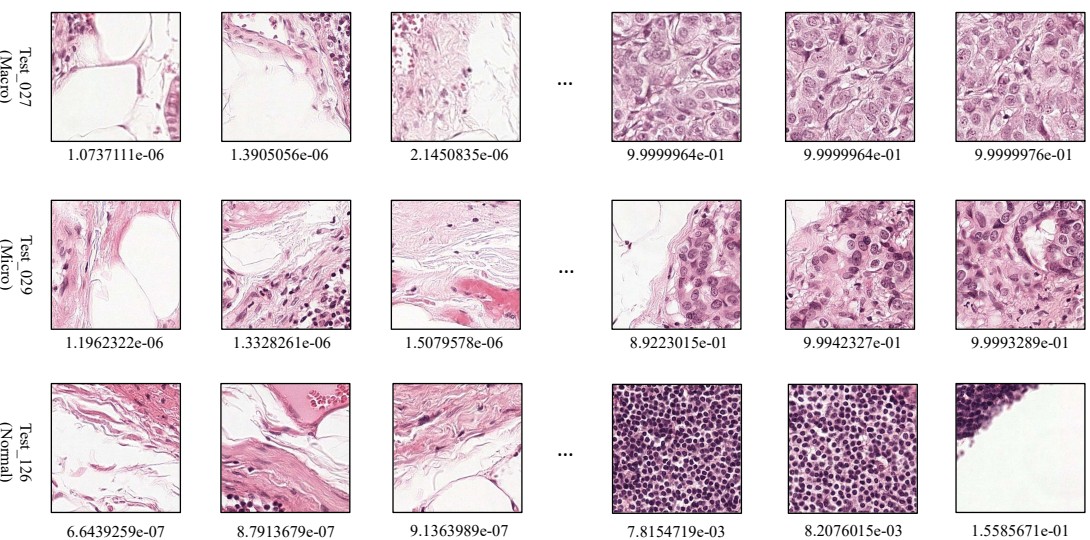

Figure 3: Visualization on Camelyon16

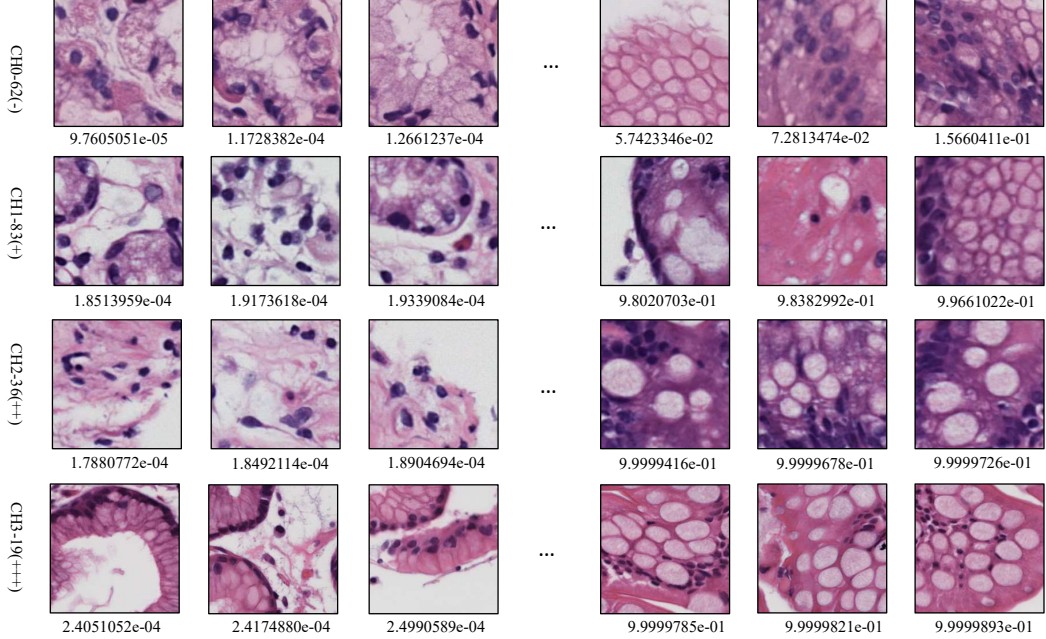

Figure 4: Visualization on BgIM

metaplasia grade 1, the projection head incorrectly assumed that negative mucosal images were positive. However, it still retrieved all patches where intestinal metaplasia actually occurred. In contrast, the projection head's judgment was more accurate for slides with intestinal metaplasia grades 2 and 3.

