# OpenReview forum: "NcIEMIL: Rethinking Decoupled Multiple Instance Learning Framework for Histopathological Slide Classification"
_MIDL.io/2024/Conference — MIDL 2024 Poster_

### Official Review · Reviewer_wvfJ · 2024-02-28

**Confidence:** 3
**Preliminary Rating:** 3
**Recommendation:** Poster
**Final Rating:** 3.5

**Summary:**

The authors tackle the whole slide image classfication task using using a multiple instance learning framework, and propose several modifications to current state-of-the-art methods to resolve some of their limitations. They provide image-level label at the feature selection stage, overcoming the limits of fully- or semi- decoupled methods. They also carefully design a way to better select and and aggregate the most relevant feature for slice classification. This allows them to outperform state-of-the-art methods on a publicly avaible dataset and on a private dataset.

**Strengths:**

- The paper is overall well-written, complete and the figures are well done and help understanding the proposed method.

- The authors have included an ablation study, which is useful for assessing the architectural choices made.

- The model that is proposed outperform baselines to which it is compared on all metrics. The evaluation is done at least on one publicly available dataset, which is good for reproductibility.

**Weaknesses:**

- It's not easy to understand the differences between the model proposed in this work and the one upon which it is based (Campanella et al., 2019).

- The comparison study with state-of-the-art method may benefit from including the reference method of (Campanella et al., 2019).

- The analysis of results, discussion and conclusions paragraphs are quite short and would benefit from being more complete.

**Detailed Comments:**

- In the discussion about the ablation study you say "We then tested different extractors [...]" but only show the results with or without ImageNet pretrained weights. What other extractor have you tried ?

- Considering the amount of instances in a given bag and the fact that in pre-training, the gradients pass through a single instance only, isn't the pre-training time very long for the encoder to extracts relevant features ?

- I felt that the paragraph 2.2, about Instance Filtering, in the Related works, is less clear than the rest and would maybe needs to be reworded to better explain how the proposed method compares with other state-of-the-art approaches.

- To complete the analysis of the results, it might be interesting to show (in the appendix for instance) some examples of patches that are selected as having the highest and lowest probabilities, in order to have a visual validation of the hypothesis that is made : "Including the latter K instances aims to implicitly construct negative samples for discriminative instances while avoiding extremes in the X data distribution."

**Justification Of Final Rating:**

I would tend to recommand accepting the paper although I am still not totally sure about the rating I give to this work. On the one hand, the method follows classical MIL workflow (feature  extractor then classification) but on the other hand the design choices gives better results than several baselines and the model is evaluated on a public dataset. The paper is overall of good quality and the authors have address the concerns that have been raised with care and details.

**Justification Of The Preliminary Rating:**

The paper is quite methodical, explaining well the locks in current architectures that it manages to remove, and achieve good results. However, it lacks clarity in its differentiation from the reference paper on which it is based, (Campanella et al., 2019). A better explanation of the additions made by the authors would enable to better assess the interest of this work.

**Questions To Address In The Rebuttal:**

- The authors are invited to make more explicit the differences between their model and the one of (Campanella et al., 2019). This would help clarify whether their paper contains significant methodological novelties, or whether it is an application of method of (Campanella et al., 2019) to a new task/database.

- If the proposed model is not exactly the same as (Campanella et al., 2019), why not include it in the methods compared in Table 1 ? This would allow to understand the value of the additions that are made compared to (Campanella et al., 2019) (if there are any).

- The performance of the proposed model is systematically superior to the baselines. However, standard errors are also quite high: have you performed statistical tests to validate that the increase in performance of your model is indeed significant ?

**Special Issue:**

No

---

> ### Author Response · Authors · 2024-03-18
>
> Thank you for recognizing our work and for your valuable suggestions. And we hope the following statements address your concerns (As the image cannot be compiled in openreview, please see <https://github.com/polyethylene16/NcIEMIL/blob/main/rebuttal/wvfJ.pdf> for details) :
>
> - **Differences between MIL-RNN [1] and our work.** Our work differs significantly from MIL-RNN. Firstly, our approach stems from addressing informational redundancies within embedding-level methods and aims to improve them. In contrast, MIL-RNN primarily focuses on instance-level methods, where techniques like RNN and random forest serve as improvements over basic aggregation functions. Secondly, while MIL-RNN primarily deals with binary classification tasks in pathology, the BgIM dataset we collected is quadruple classified. Thirdly, although both our work and MIL-RNN involve pre-training the feature aggregator, a key distinction lies in strategy during training. MIL-RNN randomly samples instances within a bag, whereas we utilize all instances, leading to a longer inference but avoiding information loss. Fourthly, our aggregator structure represents a significant innovation in itself. Finally, in terms of instance filtering, MIL-RNN selects an equal number of discriminative instances as the number of recurrences, which is influenced by the mechanism of RNN. In contrast, we adopt a different approach by filtering a larger number of non-focal instances to create bag embeddings through bidirectional sampling. To illustrate this, based on your suggestion, we appended the MIL-RNN results on both datasets as shown in the table below. And NcIEMIL proved to be more effective.
>
> | Method | CAMELYON16 ACC | CAMELYON16 AUC | CAMELYON16 F1-Score | BgIM ACC | BgIM AUC | BgIM F1-Score |
> |--------|----------------|----------------|---------------------|----------|----------|---------------|
> | MIL-RNN| $82.64_{2.17}$ | $85.02_{2.73}$ | $80.64_{1.94}$      | $53.81_{3.56}$ | $80.19_{1.86}$ | $48.55_{2.81}$ |
> | NcIEMIL| $\mathbf{86.05_{1.55}}$ | $\mathbf{89.68_{2.10}}$ | $\mathbf{85.26_{1.54}}$ | $\mathbf{85.23_{0.95}}$ | $\mathbf{95.87_{0.60}}$ | $\mathbf{81.20_{0.94}}$ |
>
> - **Significance verification.** We reported the p-values of two-sample t-tests on ACC, AUC, and F1-Score between our method and baselines respectively. $p < 0.05$ indicates that our method is significantly better than the baseline method.
>
> | Method | CAMELYON16 ACC | CAMELYON16 AUC | CAMELYON16 F1-Score | BgIM ACC | BgIM AUC | BgIM F1-Score |
> |--------|----------------|----------------|---------------------|----------|----------|---------------|
> | MIL-RNN| $0.0086$ | $0.0194$ | $0.0006$ | $5.25e-5$ | $0.0001$ | $1.91e-5$ |
> | ABMIL  | $0.0011$ | $0.0060$ | $0.0017$ | $8.51e-7$ | $7.48e-5$ | $1.95e-7$ |
> | CLAM-MB| $0.0318$ | $0.0089$ | $0.0193$ | $0.0008$ | $0.0008$ | $3.09e-5$ |
> | DSMIL  | $0.0086$ | $0.0059$ | $0.0015$ | $4.92e-5$ | $0.0001$ | $4.63e-5$ |
> | TransMIL| $0.1091$ | $0.1874$ | $0.0451$ | $0.0010$ | $0.0003$ | $0.0005$ |
> | ILRA-MIL| $0.1893$ | $0.0581$ | $0.1191$ | $0.0002$ | $0.0012$ | $0.0002$ |
>
> - **Extra extractor.** Based on your suggestion, we report the effect of self-supervised training [2] of swin-tiny as a feature extractor as follows:
>
> | Ablation item | CAMELYON16 ACC | CAMELYON16 AUC | CAMELYON16 F1-score | BgIM ACC | BgIM AUC | BgIM F1-score |
> |---------------|-----------------|-----------------|----------------------|----------|----------|---------------|
> | w/ ImageNet weight  | $85.12_{1.58}$ | $87.02_{3.87}$ | $84.13_{1.22}$ | $80.00_{1.90}$ | $92.49_{0.97}$ | $70.41_{1.05}$ |
> | w/ ctranspath weight| $85.73_{1.69}$ | $88.45_{1.61}$ | $84.84_{1.64}$ | $77.62_{4.15}$ | $92.70_{0.66}$ | $70.16_{5.87}$ |
> | NcIEMIL               | $\mathbf{86.05_{1.55}}$ | $\mathbf{89.68_{2.10}}$ | $\mathbf{85.26_{1.54}}$ | $\mathbf{85.23_{0.95}}$ | $\mathbf{95.87_{0.60}}$ | $\mathbf{81.20_{0.94}}$ |
>
> - **Time-consumption of pre-training.** We completely agree with your perspective. Indeed, the pre-training process of the feature extractor does require some time, falling somewhere between directly loading ImageNet weights and engaging in self-supervised learning. The feature extractor typically converges within 80 to 120 epochs. However, there are ways to address this drawback. For example, one approach could involve randomly sampling within the negative bags while keeping the positive bags untouched. Additionally, performing distributed computation could also help alleviate the time consumption.
>
> - **Visualization.** According to your suggestion, we present the top-3 instances and the negative top-3 instances from the 4-grades slides of BgIM and normal, micro-, and macro-metastasis slides of CAMELYON16, respectively. This section will be included in the appendix.
>
> Thanks again for your comments, and we will adjust the wording of section 2.2, although there is little relevant research on instance filtering. All additional experimental results will be added to the paper.

---

> ### Author Response · Authors · 2024-03-18
> **Reference**
>
> **Reference:**
>
> [1] Campanella, Gabriele, et al. "Clinical-grade computational pathology using weakly supervised deep learning on whole slide images." Nature medicine 25.8 (2019): 1301-1309.
>
> [2]Wang, **yue, et al. "Transformer-based unsupervised contrastive learning for histopathological image classification." Medical image analysis 81 (2022): 102559.

---

### Official Review · Reviewer_9snc · 2024-02-29

**Confidence:** 3
**Preliminary Rating:** 3
**Final Rating:** 3.5

**Summary:**

The paper proposes a novel framework called Non-crucial Information Elimination-based MIL (NcIEMIL) to improve whole slide image (WSI) classification in pathological diagnosis, particularly focusing on histopathological slide examination.

**Strengths:**

1. Authors make code public.
2. The approach is tested on 2 datasets, one of which is public.
3. The authors tackle a relevant issue.
4. The authors perform ablation studies for suggested improvements.

**Weaknesses:**

1. Normally the instance-level feature concatenation is not an issue an issue:
1.1. Either information about which areas are relevant is available priorly and filtered before the feature extractor.
1.2. If the irrelevant instances are forwarded into the aggregator the attention mechanism can filter these out by learned weights without a major computational effort.
2. Scores of baselines on Camelyon16 are much lower than reported in original papers and also other methods that used these frameworks. Why?

**Detailed Comments:**

The article is not easy to follow. For example, the sentence in the abstract uses terms that were not defined before
"We reconsider the task-specific irrelevance of most instances and the semantic redundancy of the instance representations. " Which instances? What is meant by task-specific irrelevance? What is meant by semantic redundancy? Which instance representations? The authors should improve the readability of the text.

**Justification Of Final Rating:**

The paper generally is of good quality. I still have doubts about the added benefits of the proposed changes. The authors addressed some of my comments. I modified my ranking from Borderline to Borderline accept.

**Justification Of The Preliminary Rating:**

The authors make the code public and test their method on 2 datasets. At the same time, the scores of baselines reported in this study are rather low in comparison to the ones reported in the papers. In addition, the motivation for irrelevant instances filtering is not highly convincing.

**Questions To Address In The Rebuttal:**

See weaknesses and detailed comments.

---

> ### Author Response · Authors · 2024-03-18
>
> Thank you for your invaluable feedback and comments. We are sorry that the lack of specificity in the writing confused you. We hope the following statements address your concerns (As the image cannot be compiled in openreview, please see <https://github.com/polyethylene16/NcIEMIL/blob/main/rebuttal/9snc.pdf> for details) :
>
> - **Filtering of irrelevant instances and upper bounds on the capacity of the attention mechanism.** For issue 1.1: Before feature extraction, our method is consistent with other baseline methods, which all use the OTSU algorithm to remove background patches including blank areas, fat areas, etc. While filtering of instance-level features occurs between extraction and aggregation. Conventional embedding-level methods extract high-dimensional features from patches, which will all be considered by the aggregator. However, our proposed method first sorts all patches using positive scores, and then filters out the top k ones (regarded as potential positive instances) and the negative top k ones (which are regarded as potential negative instances) as input to the aggregator, which helps reduce the noisy signal brought by too many negative instances [1]. To further prove the effectiveness of our proposed method, we consider the Camelyon16 dataset, which is a binary classification task of WSI distinguishing the presence and absence of cancer, in which the WSI with cancer is regarded as a positive bag (patches with cancer are regarded as positive instances), the non-cancer WSI is regarded as a negative bag (patches without cancer are regarded as negative instances), and the attention mechanism-based methods will give each instance a positive-related importance score. However, in practice, they will mistakenly give high scores to negative instances sometimes. As shown in the figure below. We demonstrate the attention scores assigned to instances by ABMIL and the positive scores assigned to instances by our classifier (filter) in two micro-metastatic cancer slides. Negative instances are highlighted with black frames, while positive ones are marked in red. Instances are sorted in descending order from left to right by attention. For ease of observation, the attention is scaled by the softmax operator. In Test_10, three negative instances are assigned the second to fourth highest attention, but our method successfully eliminated these negative examples and correctly classified the WSI. In Test_84, negative instances containing a large number of lymphocytes received high attention in ABMIL, while positive instances were not in the high attention list, resulting in misclassification. Our method tended to eliminate these negative instances.
>
> ![Figure 1](1.pdf)
>
> For issue 1.2: As shown in the following figure, for Test_84, irrelevant instances (negative instances) are fed into the aggregator of ABMIL, but the attention mechanism considers them relevant and leads to misclassification. We also show the results of ranking the positive scores generated according to our filter, where a large number of positive instances remain in the higher ranks despite the lower scores. These relevant instances are encapsulated in the new pseudo-bag.
>
> ![Figure 2](2.pdf)
>
> - **Results of baseline methods.** We do not reproduce the results reported in the original article using the baseline model on the CAMELYON16 dataset for several reasons. Firstly, our experimentation involved a 5-fold cross-validation, which differed from that used in the baseline article. To enhance the credibility of our study, we provide detailed information regarding the division of the 5 folds, which can be accessed at [this link](https://github.com/polyethylene16/NcIEMIL/tree/main/fold). Secondly, despite employing the same preprocessing strategy, we did not generate the same number of patches as reported in the other articles. Finally, we did not employ any additional techniques during training; for instance, in the comments section of [this link](https://github.com/szc19990412/TransMIL/issues/25), a commenter only achieved slightly lower results by employing a trick (early-stop) not mentioned in the original article. During experiments, We directly used the training code provided by the authors and conducted with our partitioned dataset.
>
>
> - **Modifications of the abstract.** We have updated the abstract to address poor readability and replace "We reconsider the task-specific irrelevance of most instances and the semantic redundancy of the instance representations" with "This exacerbates the information bottleneck, leading to instance representations in a high-rank space that contains semantic redundancy compared to the potential low-rank category space of instances. Additionally, most negative instances are also independent of the positive properties of the bag."
>
> In response to the poor readability of this paper, we will adjust the paragraphs, including the abstract, in the final version of the paper. Thanks again for your comments!

---

> > ### Author Response · Authors · 2024-03-18
> > **Reference**
> >
> > **Reference**:
> >
> > [1] Yan, Renao, et al. "Shapley Values-enabled Progressive Pseudo Bag Augmentation for Whole Slide Image Classification." arv preprint arv:2312.05490 (2023).

---

### Official Review · Reviewer_4Xqy · 2024-03-02

**Confidence:** 4
**Preliminary Rating:** 4
**Final Rating:** 4

**Summary:**

The paper presents a multiple instance learning based WSI classification method. Overall pipeline consists of:
+ feature extraction
+ feature selection based on probability score
+ feature aggregation based on spatial and channel attention mechanism to predict the level of the image

Results are provided on two datasets, and the proposed MIL approach performs better the compared MIL approaches.

**Strengths:**

Although overall structure of the methodology is not unique (training a feature extraction network in the first stage, utilising this network in the second stage for feature extraction and utilising attention-based pooling in unison with RNN, LSTM, etc., for label prediction has been explored exhaustively in the existing MIL work), authors have redesigned individual components to:
+ Reduce bag size
+ Channel weightage

These modification simplifies the computation and improves the performance.
The paper is also well written and mostly easy to follow.

**Weaknesses:**

Writing:
+ The writing can be improved in the *methodology* section
     + *Figure-1* should have a more elaborate caption (providing a more detailed description)
     + $e_s$ (embeddings), $\textbf{u}$, and $\textbf{v}$ should be explained in detail and how are they derived
     + The dimensions of the above should also be included for better readability

Results:
 + The results with non-MIL based approach should also be included
+  The ablation study for $k$ can also be included

**Detailed Comments:**

Please refer to *strengths* and *weaknesses* sections

**Justification Of Final Rating:**

The authors have addressed the previous, and have include all the required results (such as comparison to non-MIL based methods and ablation study of the k). I retain the previous rating, favouring the acceptance.

**Justification Of The Preliminary Rating:**

The decision is based on the methodology and provided results alongwith the ablation study. The proposed approach of bag reduction can be included in the existing MIL approaches. Similarly, the proposed attention based aggregation can be utilised in the existing approaches.

**Questions To Address In The Rebuttal:**

Please refer to the *weaknesses* section

---

> ### Author Response · Authors · 2024-03-18
>
> Thank you for recognizing our work and for your valuable suggestions. And we quite agree with you, our point-to-point response is as follows (As the image cannot be compiled in openreview, please see <https://github.com/polyethylene16/NcIEMIL/blob/main/rebuttal/4Xqy.pdf> for details.) :
>
> - **Writing improvement.** In response to your suggestion, we have described Fig. [fig1] in more detail as follows. Furthermore, exhaustive presentations of the sources and dimensions of embeddings, such as **u** and **v**, will be provided. For example, "where **u** = $\in \mathbb{R}^{2K \times d}$ denotes the bag, obtained by concatenating instance embeddings, with *d* being the embedding dimension", "**v**  $\in \mathbb{R}^{1 \times d}$ is a learnable embedding that represents a virtual instance used for classification" and "Then the aggregated **v'** is extracted from the spatial dimension of **u'**  $\in \mathbb{R}^{(1 + 2K) \times d}$. These changes will be updated in the final version.
>
> ![Overview of NcIEMIL. (a) Instances are fed into the extractor and projection head on a bag basis, and the instance with the highest probability is involved as a representative of its corresponding bag. (b) Instances within a bag are re-ranked, and bi-directional sampling is employed to generate bag embeddings. (c) A hybrid attention-based aggregator, comprising parallel spatial and channel attention mechanisms, is employed to mitigate information redundancy.](1.pdf)
>
> - **Additional comparison experiments.** Following your suggestions, we have incorporated two additional methods into our comparative experiment. The naive image-wise approach involves directly classifying downsampled WSI thumbnails using a classification network, akin to the approach in \[1\]. The fully-supervised approach, on the other hand, uses a batch of patches with real labels to train a classification network. In CAMELYON16, we extracted positive and negative patches from within and without cancerous regions respectively, according to the pixel-level annotations provided officially. Regarding BgIM, we randomly selected 2,000 preprocessed patches and annotated them for fully supervised training. The results of the slides were obtained by the mean value of all instances. The results are presented in the following table and will be incorporated into Tab. 1 of the original paper. To be fair, swin-tiny was chosen for all the above classification networks to maintain consistency.
>
> | Method           | CAMELYON16  ACC | CAMELYON16 AUC | CAMELYON16 F1-Score | BgIM ACC | BgIM AUC | BgIM F1-Score |
> |------------------|----------------|----------------|----------------------|----------|----------|----------------|
> | Naive            | $62.40_{1.42}$          | $62.16_{1.40}$          | $45.15_{5.28}$                | $52.98_{3.32}$    | $76.58_{1.28}$    | $44.27_{4.44}$          |
> | Fully-supervised| $ \color{red}{91.32_{1.99}} $     | $ \color{red}{94.93_{0.87}} $     | $ \color{red}{90.73_{2.12}} $           | $ \color{red}{91.91_{1.17}} $| $ \color{red}{98.72_{0.33}} $| $ \color{red}{91.57_{2.61}} $     |
> | NcIEMIL          | $\mathbf{86.05_{1.55}}$      | $\mathbf{89.68_{2.10}}$     | $\mathbf{85.26_{1.54}}$           | $\mathbf{85.23_{0.95}}$| $\mathbf{95.87_{0.60}}$| $\mathbf{81.20_{0.94}}$      |
>
> - **Additional ablation experiments.** We also performed ablation experiments on *K*. The results are shown in the following table. Specifically, we introduced two values for *K*: a small *K* (*K*=128 for CAMELYON16 and *K*=32 for BgIM) and a medium *K* (*K*=288 for CAMELYON16 and *K*=72 for BgIM), and assessed their impacts on the results. The reason for not selecting a value larger than *K* in the original article (*K*=512 for CAMELYON16 and *K*=128 for BgIM) is because of the restricted number of instances in the smallest bag within the dataset.
>
> | Ablation item   | CAMELYON16 ACC | CAMELYON16 AUC | CAMELYON16 F1-score | BgIM ACC | BgIM AUC | BgIM F1-score |
> |-----------------|-----------------|-----------------|----------------------|----------|----------|----------------|
> | w/ small *K*    | $85.75_{1.84}  $         | $\mathbf{90.06_{1.87}}$     | $84.87_{1.68}  $              | $83.33_{1.51}  $  | $95.21_{0.50}  $  | $80.09_{2.14}  $        |
> | w/ medium *K*   | $85.36_{1.49}  $         | $89.46_{1.30}   $        | $84.01_{1.30}  $              | $84.28_{1.90}   $ | $94.90_{0.83}  $  | $80.00_{1.95}   $       |
> | NcIEMIL         | $\mathbf{86.05_{1.55}}$      | $89.68_{2.10} $          | $\mathbf{85.26_{1.54}}$           | $\mathbf{85.23_{0.95}}$| $\mathbf{95.87_{0.60}}$| $\mathbf{81.20_{0.94}}$      |
>
> [1] Chen, Chi-Long, et al. "An annotation-free whole-slide training approach to pathological classification of lung cancer types using deep learning." Nature communications 12.1 (2021): 1193.

---

### Meta-Review · Area_Chair_H5F2 · 2024-04-04

**Recommendation:** Accept (Poster)
**Confidence:** 5

**Metareview:**

The paper presents a multiple instance learning (MIL) based approach for whole slide image (WSI) classification in pathological diagnosis, introducing a novel framework called Non-crucial Information Elimination-based MIL (NcIEMIL). The proposed method involves feature extraction, selection based on probability score, and aggregation using spatial and channel attention mechanisms to predict image labels. Results on two datasets demonstrate improved performance compared to existing MIL approaches.

Overall, the paper presents a promising approach for WSI classification using MIL, with improvements in feature selection and aggregation mechanisms. Addressing the weaknesses mentioned by reviewers will strengthen the paper's contribution to the field.

---

### Decision · Program_Chairs · 2024-04-06

Accept (Poster)